# Ouroboros: On Accelerating Training of Transformer-Based Language Models

**Qian Yang[1]\***, **Zhouyuan Huo[2]**, **Wenlin Wang[1]**, **Heng Huang[2]**, **Lawrence Carin[1]**

Department of Electrical and Computer Engineering
[1] Duke University    [2] University of Pittsburgh
`qian.yang@duke.edu`

## Abstract

Language models are essential for natural language processing (NLP) tasks, such as machine translation and text summarization. Remarkable performance has been demonstrated recently across many NLP domains via a Transformer-based language model with over a billion parameters, verifying the benefits of model size. Model parallelism is required if a model is too large to fit in a single computing device. Current methods for model parallelism either suffer from backward locking in backpropagation or are not applicable to language models. We propose the first model-parallel algorithm that speeds the training of Transformer-based language models. We also prove that our proposed algorithm is guaranteed to converge to critical points for non-convex problems. Extensive experiments on Transformer and Transformer-XL language models demonstrate that the proposed algorithm obtains a much faster speedup beyond data parallelism, with comparable or better accuracy. Code to reproduce experiments is to be found at `https://github.com/LaraQianYang/Ouroboros`.

## 1   Introduction

Natural language processing (NLP) tasks, such as machine translation [1, 2, 3, 4, 5], text summarization [6, 7, 8, 9, 10], or paraphrase generation [11, 12, 13] have achieved great success with the development of neural networks. It has been demonstrated recently that Transformer networks obtain superior performance [14, 15, 16] relative to recurrent neural networks or convolutional neural networks. BERT [17] trains a deep bidirectional Transformer with 340M parameters and obtains state-of-the-art results on 11 NLP tasks. Recently, OpenAI GPT-2 [18], which is a Transformer-based language model with 1.5B parameters, achieves state-of-the-art results on 7 out of 8 tested language modeling datasets, presenting impressive performance across many domains and datasets. Empirical results demonstrate the superiority of Transformer networks and show that a larger model tends to yield better performance. However, when a model is so large that it has to be allocated on multiple GPUs, data parallelism over these GPUs is not applicable because it requires each GPU to have one copy of the whole model. Meanwhile, model parallelization is still an open question when the model is too large to fit in a single device when training.

When a model becomes too large to fit on a single computing device, the simplest solution is to distribute model layers across multiple devices. In [19], the authors parallelize the model by splitting filters or parameters of a layer across multiple GPUs. However, both of these methods suffer from backward locking of the backpropagation algorithm, and cannot parallelize the computations between layers. Backward locking denotes that the backpropagation algorithm requires gradients to be

---

computed from top layers to bottom layers sequentially. When networks are very deep, all other devices are idle when the backpropagation computation is performed on one device. Jaderberg et al. [20] proposes Decoupled Neural Interface to remove backward locking, by employing additional neural networks to approximate error gradients. However, this approach works poorly on deep neural networks [21]. In [21], the authors use stale gradients in previous computations and successfully accelerate the training of deep networks like ResNet110. Subsequently, Huo et al. [22] revises the memory issue in [21] and obtains better generalization error. Both of these methods can only work on feed-forward networks that are separable between layers. However, neither approach can parallelize Transformer-based language models, because the shared embeddings make the networks non-separable.

To address the above challenges, we make the following contributions. $(i)$ We present the first model-parallel algorithm to parallelize the training of Transformer-based language models, going beyond data parallelism. $(ii)$ The convergence rate of the proposed algorithm is analyzed, and it is proven that it is guaranteed to converge to critical points for non-convex problems. $(iii)$ We evaluate the proposed algorithm in training two Transformer-based language models, and experimental results verify our theoretical analysis, demonstrating convergence much faster than previous methods with comparable or better accuracy. The source code will be made publicly accessible to encourage further research.

## 2    Preliminary and Related Works

Self-attention architectures like the Transformer [14] have recently become popular for language modeling [15, 16, 17, 18]. Consider training a Transformer-based language model with $L$ layers. We may represent the computations in the network as follows:

$$
\begin{align}
h_1 &= F_1(h_0; w_1, V_i), \tag{1}\\
h_l &= F_l(h_{l-1}; w_l), \qquad \forall l \in \{2, ..., L-1\}, \tag{2}\\
h_L &= F_L(h_{L-1}; w_L, V_o), \tag{3}
\end{align}
$$

where $h_{l-1}$ denotes the input of layer $l$, $F_l(\cdot; w_l)$ denotes the computation of layer $l$ with weight $w_l$, $V_i$ is the input embedding, and $V_o$ is the output projection. In particular, $h_0$ denotes the input data $x$, and $h_L = F(x; \tilde{w})$ represents the output of the network. For the sake of performance, $V_i$ and $V_o$ are typically tied in language modeling or machine translation tasks, so that $V = V_i = V_o$ [23, 24]. Defining network weights $w = [w_1, w_2, ..., w_L]$, embedding layer $V$ and $\tilde{w} = [w, V]$, the loss function for language modeling can be represented as:

$$
\min_{\tilde{w}} f(F(x; \tilde{w}), y), \tag{4}
$$

where $y$ denotes the target. In the following context, we use $f(\tilde{w})$ for simplicity.

### 2.1    Gradient-Based Method

Gradient-based methods are widely employed for training deep neural networks, with important stochastic gradient descent (SGD) [25] examples including AdaGrad [26], RMSProp [27], Adam [28] and AdamW [29]. With SGD, the weights of the network are updated as:

$$
w_l^{t+1} = w_l^t - \gamma_t \nabla f_{l,x_{i(t)}}(\tilde{w}^t) \quad \text{and} \quad V^{t+1} = V^t - \gamma_t \nabla f_{V,x_{i(t)}}(\tilde{w}^t), \tag{5}
$$

for any $l \in \{1, ..., L\}$, where $\gamma_t$ is the stepsize, $i(t)$ represents data index at iteration $t$, and $\nabla f_{l,x_{i(t)}}(\tilde{w}^t)$ is the gradient of the loss function (4) with respect to the weights at layer $l$ and data sample $x_{i(t)}$.

### 2.2    Backpropagation

If the loss functions are differentiable, the gradients of network parameters can be computed using the backpropagation algorithm [30]. The backpropagation algorithm consists of two passes of the network, forward computation and backward computation. In the forward computation, activations of all layers are calculated from $l = 1$ to $L$ following equations (1), (2) and (3). In the backward

**Algorithm 1** Ouroboros + SGD

**Require:**
  Initial weights $w^0 = [w^0_{\mathcal{G}(1)}, ..., w^0_{\mathcal{G}(K)}]$;
  Initial word embedding $V_i^0 = V_o^0$;
  Stepsize sequence $\{\gamma_t\}$;
1: **for** $t = 0, 1, 2, \ldots, T-1$ **do**
2:   **for** $k = 1, \ldots, K$ **in parallel do**
3:     Compute delayed gradient $g_k^t$ for module $k$ following (8);
4:     Compute mixed gradient $g_V^t$ for embedding layer following (9);
5:     Update weights and embedding layer following SGD:

$$w^{t+1}_{\mathcal{G}(k)} = w^t_{\mathcal{G}(k)} - \gamma_t \cdot g_k^t;$$
$$V_i^{t+1} = V_o^{t+1} = V_i^t - \gamma_t \cdot g_V^t;$$

6:   **end for**
7: **end for**
8: Output $w^s$, $V_i^s$ and $V_o^s$ randomly from $\{w^t\}_{t=0}^{T-1}$, $\{V_i^t\}_{t=0}^{T-1}$ and $\{V_o^t\}_{t=0}^{T-1}$.

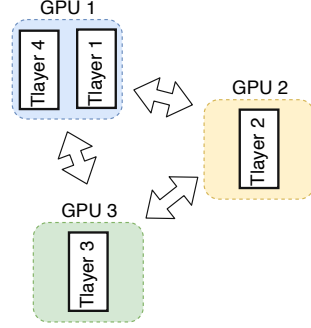

Figure 1: Communication between GPUs of the proposed Ouroboros algorithm. The first and last module of a transformer-based language model is located on the same device.

computation, we apply the chain rule and propagate error gradients repeatedly through the network, from the output layer $l = L$ to the input layer $l = 1$:

$$\frac{\partial f(\tilde{w}^t)}{\partial w_l^t} = \frac{\partial f(\tilde{w}^t)}{\partial h_l^t} \times \frac{\partial h_l^t}{\partial w_l^t} \quad \text{and} \quad \frac{\partial f(\tilde{w}^t)}{\partial h_{l-1}^t} = \frac{\partial f(\tilde{w}^t)}{\partial h_l^t} \times \frac{\partial h_l^t}{\partial h_{l-1}^t}, \tag{6}$$

where $\tilde{w} = [w, V]$, and $\nabla f_{l,x_{i(t)}}(\tilde{w}^t) = \frac{\partial f(\tilde{w}^t)}{\partial w_l^t}$. For Transformer-based language models, the gradient with respect to the input embedding and output projection layer are computed as:

$$\frac{\partial f(\tilde{w}^t)}{\partial V_i} = \frac{\partial f(\tilde{w}^t)}{\partial h_1^t} \times \frac{\partial h_1^t}{\partial V_i} \quad \text{and} \quad \frac{\partial f(\tilde{w}^t)}{\partial V_o} = \frac{\partial f(\tilde{w}^t)}{\partial h_L^t} \times \frac{\partial h_L^t}{\partial V_o}. \tag{7}$$

From (6), it is evident that the computation in layer $l$ is dependent on the error gradient $\frac{\partial f(\tilde{w}^t)}{\partial h_l^t}$ from layer $l + 1$. Therefore, the sequential chain rule constrains all layers from updating before receiving error gradients from the dependent layers. When Transformer-based networks are very deep and computations in each layer are significant, breaking such a sequential chain rule to accelerate the training presents a challenge.

## 3 Accelerating Training of Transformer-Based Language Models

We propose the first model-parallel algorithm that can speed up the training of Transformer-based language models. We then take stochastic gradient descent as an example to verify that our algorithm is easy to work with any gradient-based method.

### 3.1 Ouroboros Algorithm

We split an $L$-layer network into $K$ modules so that the weights of the network are divided into $K$ groups and each group is placed on a GPU. Therefore, we have $w = [w_{\mathcal{G}(1)}, w_{\mathcal{G}(2)}, ..., w_{\mathcal{G}(K)}]$ where $\mathcal{G}(k)$ denotes layer indices in group $k$. We again denote $V_i$ and $V_o$ as the input embedding and output projection, at the first and last module of the network. In [23], it is shown that shared embedding always has better performance than not sharing for a language model and machine translation, where $V_i$ and $V_o$ are tied and $V_i = V_o$. In the following context, we let $V = [V_i, V_o]$. Because of this, the first module and the last module must be placed on the same device, visualized in Figure 1. Our model is connected end-to-end and shrinks like a snake when grouping, so we name it "Ouroboros."

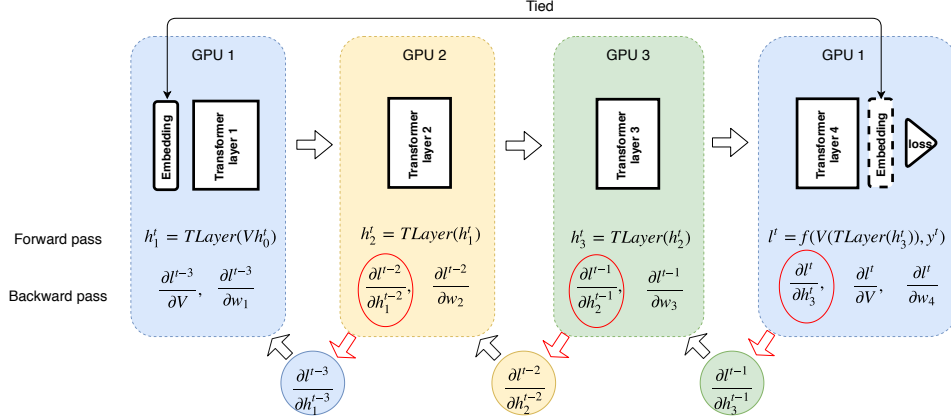

Figure 2: Forward and backward computation of the proposed algorithm. We split a Transformer-based language model into four modules and allocate them into three GPUs, where the first and the last module are placed on the same GPU. In the figure, $h$ denotes activations, $w$ denotes weights, and $V$ represents embedding layers. $TLayler$ represents Transformer layer. The input embedding and output projection are tied together.

In the backward computation of the backpropagation algorithm, the computations of Module 1 are dependent on the computations of the later modules. In our Ouroboros algorithm, at each iteration all modules are independent of each other, by using delayed gradients. Let $\tilde{w} = [w, V]$, the gradient of weights in $\mathcal{G}(k)$ is

$$\nabla f_{\mathcal{G}(k), x_{i(t-K+k)}}\left(\tilde{w}^{t-K+k}\right) = \sum_{l \in \mathcal{G}(k)} \frac{\partial f_{x_{i(t-K+k)}}(\tilde{w}^{t-K+k})}{\partial w_l^{t-K+k}}, \text{ if } t - K + k \geq 0, \qquad (8)$$

or 0 otherwise for any $k \in \{1, 2, ..., K\}$. The gradient of $V$ is the average of the gradients of output projection and input embedding:

$$\nabla f_{V, x_{i(t)}}(\tilde{w}^t) = \frac{1}{2} \nabla f_{V_o, x_{i(t)}}\left(\tilde{w}^t\right) + \frac{1}{2} \nabla f_{V_i, x_{i(t-K+1)}}\left(\tilde{w}^{t-K+1}\right) = \frac{1}{2}\left(\frac{\partial f(\tilde{w}^t)}{\partial V_o^t} + \frac{\partial f(\tilde{w}^{t-K+1})}{\partial V_i^{t-K+1}}\right), \quad (9)$$

otherwise 0 if $t - K + 1 < 0$. In the proposed algorithm, the backward computation in module $k$ is always one time step behind module $k + 1$. Therefore, the computations in all modules can be parallelized. In Figure 2, we visualize the procedure of the Ouroboros algorithm, optimizing a Transformer-based language model with four modules.

**Memory Consumption.** In the Ouroboros algorithm, we need to store stale gradients of all layers, which may be memory demanding. We follow [31] and only store the input of each GPU. Required activations and gradients are recomputed in the backward pass. Therefore, the extra memory consumption is negligible, which is only dependent on the number of GPUs.

### 3.2 Gradient-Based Method with Ouroboros

After obtaining gradients of the loss function with respect to the weights of the model, we can apply these gradients to gradient-based methods. We consider the procedures of SGD as an example. Letting $g_k^t$ and $g_V^t$ represent the gradients of module $k$ and embedding $V$ at iteration $t$, we can update model weights and embeddings following SGD:

$$w_{\mathcal{G}(k)}^{t+1} = w_{\mathcal{G}(k)}^t - \gamma_t \cdot g_k^t; \qquad (10)$$

$$V_i^{t+1} = V_o^{t+1} = V_i^t - \gamma_t \cdot g_V^t, \qquad (11)$$

where $\gamma_t$ denotes the stepsize. We summarize Ouroboros with SGD in Algorithm 1. In the next section, we analyze the convergence rate of Algorithm 1, which is the basis of analysis for other variants of SGD.

# 4 Convergence Analysis

We prove Algorithm 1 is guaranteed to converge to critical points for non-convex problems. Results show that it admits a similar convergence rate to vanilla SGD. Detailed proofs are in the supplementary material. At first, we make two commonly used assumptions following [32]:

**Assumption 1** *(Lipschitz-continuous gradient)* *The gradient of $f(w)$ is Lipschitz continuous with Lipschitz constant $L > 0$, such that for any $w, v$, it is satisfied that:*

$$\|\nabla f(w) - \nabla f(v)\|_2 \leq L\|w - v\|_2. \tag{12}$$

**Assumption 2** *(Bounded variance)* *We assume the second moment of the stochastic gradient is upper bounded, such that there exists constant $M \geq 0$, for any sample $x_i$ and for any $w$:*

$$\|\nabla f_{x_i}(w)\|_2^2 \leq M. \tag{13}$$

*Because of the variance equation $\mathbb{E}\|\nabla f_{x_i}(w) - \nabla f(w)\|_2^2 = \mathbb{E}\|\nabla f_{x_i}(w)\|_2^2 - \|\nabla f(w)\|_2^2$, the following inequality is also satisfied:*

$$\|\nabla f_{x_i}(w) - \mathbb{E}\left[\nabla f_{x_i}(w)\right]\|_2^2 \leq M. \tag{14}$$

Under Assumptions 1 and 2, we obtain Lemma 1 about iterations of the objective functions.

**Lemma 1** *With Assumptions 1 and 2, let $\sigma := \max_t \frac{\gamma_{\max\{0, t-K+1\}}}{\gamma_t}$ and $M_K = (K + \frac{3}{4})M + \sigma(\frac{K^2}{2} + K^3)(K + 4)M$. For all $t \in \mathbb{N}$, the iterations in Algorithm 1 satisfy the inequality*

$$\mathbb{E}\left[f(w^{t+1})\right] - f(w^t) \leq -\frac{\gamma_t}{2}\left\|\nabla f(w^t)\right\|_2^2 + \gamma_t^2 L M_K. \tag{15}$$

From Lemma 1, we observe that the expected decrease of the objective function is controlled by the stepsize $\gamma_t$ and $M_K$. Therefore, we can guarantee that the values of objective functions are decreasing as long as the stepsizes $\gamma_t$ are small enough, such that the right-hand side of (15) is less than zero. Based on Lemma 1, we analyze the convergence guarantee of Algorithm 1.

## 4.1 Fixed Stepsize $\gamma_t$

We first analyze the convergence for Algorithm 1 when $\gamma_t$ is fixed, and prove that the learned model will converge sub-linearly to the neighborhood of the critical points.

**Theorem 1** *With Assumptions 1 and 2, and the fixed stepsize sequence $\{\gamma_t\}$ satisfying $\gamma_t = \gamma$ and $\gamma L \leq 1, \forall t \in \{0, 1, ..., T-1\}$, let $w^*$ be the optimal solution to $f(w)$. The output of Algorithm 1 satisfies:*

$$\frac{1}{T}\sum_{t=0}^{T-1}\mathbb{E}\left\|\nabla f(w^t)\right\|_2^2 \leq \frac{2\left(f(w^0) - f(w^*)\right)}{\gamma T} + 2\gamma L M_K, \tag{16}$$

*and $M_K = (K + \frac{3}{4})M + (\frac{K^2}{2} + K^3)(K + 4)M$.*

According to Theorem 1, the average norm of gradients can converge to the neighborhood of critical points. As $T \to \infty$, it is also upper bounded by $2\gamma L M_K$.

**Remark 1** *With Assumptions 1 and 2, and following notation in Theorem 1, let $\gamma = \sqrt{\frac{f(w^0) - f(w^*)}{TLM_K}}$. Then $\frac{1}{T}\sum_{t=0}^{T-1}\mathbb{E}\left\|\nabla f(w^t)\right\|_2^2 \leq 4\sqrt{\frac{(f(w^0) - f(w^*))LM_K}{T}}$.*

According to above analysis, we know that Algorithm 1 admits a convergence rate of $O(1/\sqrt{T})$ for non-convex problems, which is similar to the result of SGD [32].

## 4.2 Diminishing Stepsize $\gamma_t$

We prove that Algorithm 1 with diminishing stepsizes can guarantee the convergence to critical points for non-convex problems.

**Theorem 2** *With Assumptions 1 and 2, and the diminishing stepsize sequence $\{\gamma_t\}$ satisfying $\gamma_t = \frac{\gamma_0}{1+t}$, $\gamma_t L \leq 1, \forall t \in \{0, 1, ..., T-1\}$, assume $w^*$ to be the optimal solution to $f(w)$, and let $\sigma = K$ such that $M_K = (K + \frac{3}{4})M + (\frac{K^3}{2} + K^4)(K + 4)M$. Setting $\Gamma_T = \sum\limits_{t=0}^{T-1} \gamma_t$, then the output of Algorithm 1 satisfies*

$$\frac{1}{\Gamma_T} \sum_{t=0}^{T-1} \gamma_t \mathbb{E} \left\| \nabla f(w^t) \right\|_2^2 \quad \leq \quad \frac{2\left(f(w^0) - f(w^*)\right)}{\Gamma_T} + \frac{2 \sum\limits_{t=0}^{T-1} \gamma_t^2 L M_K}{\Gamma_T}$$

Since $\gamma_t = \frac{\gamma_0}{t+1}$, the following inequalities are satisfied:

$$\lim_{T \to \infty} \sum_{t=0}^{T-1} \gamma_t = \infty \quad \text{and} \quad \lim_{T \to \infty} \sum_{t=0}^{T-1} \gamma_t^2 < \infty. \tag{17}$$

Therefore, according to Theorem 2, when $T \to \infty$, the right-hand side of (17) converges to 0.

**Remark 2** *Suppose $w^s$ is chosen randomly from $\{w^t\}_{t=0}^{T-1}$ with probabilities proportional to $\{\gamma_t\}_{t=0}^{T-1}$. According to Theorem 2, we can prove that Algorithm 1 guarantees convergence to critical points for the non-convex problem:* $\lim\limits_{s \to \infty} \mathbb{E}\|\nabla f(w^s)\|_2^2 = 0$.

## 5 Experimental Setup

We evaluate the proposed method by training two Transformer-based language models. When the model is too large to be fit in a single GPU, its layers have to be distributed across multiple GPUs. In this case, data parallelism over multiple GPUs does not work because it requires that each GPU has one copy of the whole model. Mini-batch computation in one GPU is regarded as the data parallelism in this paper. By simulating this case, we distribute layers of a model across $K$ GPUs. Experimental results demonstrate that the proposed method obtains further speedup beyond data parallelism.

### 5.1 Datasets

Following [16], three publicly available datasets are used for training and evaluation: ($i$) enwiki8, containing 100M bytes of unprocessed Wikipedia text [33]; ($ii$) text8, containing 100M processed lower-case Wikipedia characters and removing any character other than the 26 letters $a$ through $z$, and space [33]; and ($iii$) WikiText-103, the largest available word-level language modeling benchmark with long-term dependency [34]. All training datasets are preprocessed following [16].

### 5.2 Training Details

Our implementation is based on Transformer-XL[2] using PyTorch. All experiments are performed on a machine with $4\times$TESLA V100 GPUs. Parallelization between modules is handled via the subprocess library in Python3. We use two language models in the paper: a 12-layer Transformer (44M parameters) [15] and Transformer-XL (41M parameters) [16]. In all experiments, we split a Transformer-based language model into $K$ modules and allocate them sequentially onto $K$ GPUs (backpropagation algorithm) or $K - 1$ GPUs (Ouroboros). Due to the limited resources, we validate our proposed algorithm by varying $K$ from 3 to 5. According to [16], we use the Adam optimizer, where $\beta_1 = 0.9$, $\beta_2 = 0.999$ and $\varepsilon = 1e^{-8}$ [28]. For comparison, we use Ouroboros+Adam (see Appendix) in the experiments. The learning rate is set to be $0.00025$ and it decreases following a cosine learning rate schedule [35].

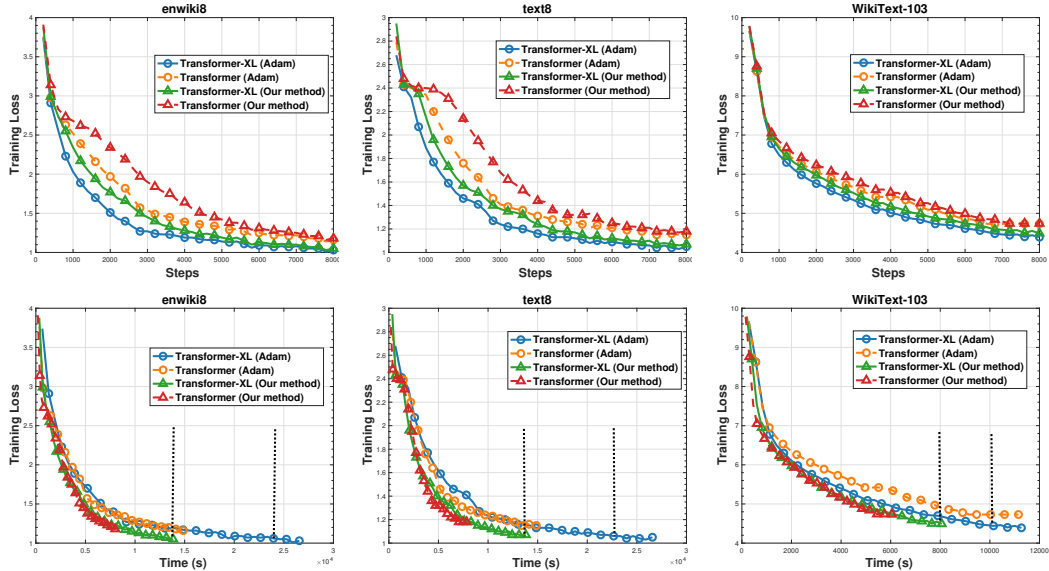

Figure 3: Convergence of the methods, regarding steps and computational time. We evaluate our algorithm on both Transformer and Transformer-XL language models.

| Dataset | Transformer | | Transformer-XL | |
|---|---|---|---|---|
| | Adam | Ouroboros + Adam | Adam | Ouroboros + Adam |
| enwiki8 | **1.11** | 1.12 | 1.06 | **1.05** |
| text8 | **1.18** | **1.18** | 1.15 | **1.13** |
| WikiText-103 | 28.32 | **28.29** | **24.00** | 24.10 |

Table 1: Comparison of Test bpc (Bit per Character) or Test PPL. We use the metric bpc on the enwiki8 and text8 datasets, and PPL on the WikiText-103 dataset. Our algorithm can achieve speedup with comparable or better performance.

**Warm-up Training.** In the early stages of training, stale gradient information may affect the convergence of Ouroboros. Following [36], we use a gradual warm-up approach in all experiments. This avoids a sudden increase of the learning rate, decreasing the error caused by stale gradients. In all experiments, we set the warm-up step to be 5000. After the warm-up, we use the cosine learning rate schedule.

**Repeatable Dropout.** According to [37], dropout ignores weights in a fully-connected layer independently and randomly, with a given probability. Ouroboros allows modules to compute gradients in parallel, in different time-stamps. To compute gradients with the input from time-stamp $t - K + k$, we need to recompute activations $h_{\mathcal{G}(k)}^{t-K+k}$ in module $k$. However, randomness in the dropout layer prevents recovering previous activations accurately. Consequently, we propose to store the input of each module as well as a random seed. Therefore, before computing activations, we initialize the random number generator in GPU with the stored seed.

### 5.3 Evaluation Metric

To evaluate the convergence rate of the proposed algorithm, we compare training loss regarding steps and computational time. We evaluate the final performance of the trained model by computing the bpc score on test data of enwiki8 and test8 datasets, and PPL score on the test data of WikiText-103.

## 6 Experimental Results

We show that our Ouroboros algorithm parallelizes the previous sequential backpropagation, and obtains a much faster speedup beyond data parallelism, without loss of accuracy. We also perform

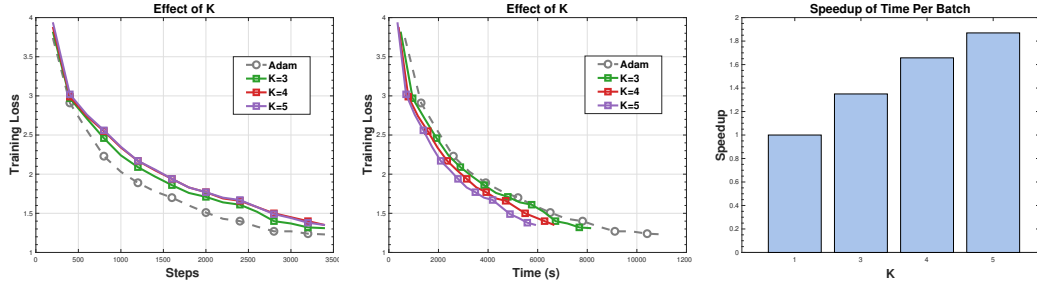

Figure 4: Convergence of training loss regarding steps and computational time, when we vary modules $K$. Speedup of computational time per batch in the right figure. Experiments are performed to train Transformer-XL on enwiki dataset

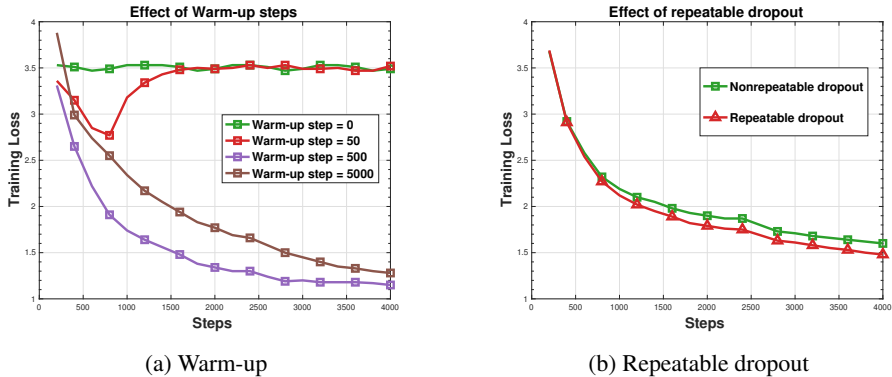

(a) Warm-up

(b) Repeatable dropout

Figure 5: Ablation study on the effect of warm-up (Figure 5a) and repeatable dropout (Figure 5b). Experiments are performed to train Transformer-XL on enwiki dataset.

ablation studies to analyze the necessity of the proposed training techniques. All figures are plotted using the average of $5$ runs.

## 6.1 Convergence Comparisons

The proposed method is evaluated by optimizing two Transformer-based language models, Transformer [15] and Transformer-XL [16]. For the enwiki and text8 datasets, we use $12$-layer models, and for WikiText-103 dataset, we use $16$-layer models. We visualize the convergence of training loss regarding steps and computational time in Figure 3. The convergence rate of our algorithm and the alternative methods are very close. This verifies our theoretical analysis that the proposed algorithm converges to critical points with a rate of $O(1/T)$. Secondly, our algorithm is much faster than alternative methods. In Table 1, we compare PPL or bpc of the methods. Experimental results show that our algorithm obtains comparable or sometimes better performance.

## 6.2 Distributed Speedup

We further evaluate our algorithm by varying $K$ from $3$ to $5$ and visualize experimental results in Figure 4. We allocate $K$ modules on $K - 1$ GPUs. Note that $(i)$ increasing the number of modules may affect the convergence regarding steps, consistent with our theoretical analysis; and $(ii)$ more speedup will be obtained when the networks are deeper. It is an ideal case to obtain linear speedup, using $K\times$ machines to achieve $K\times$ speedup regarding time. However, it is impossible to achieve even for data parallelism. The goal of our method is to guarantee that there is no idle machines during the training and fully utilize all computing resources. Besides, it is also easy to combine our method with data parallelism to obtain further speedup.

### 6.3 Ablation Studies

**The Effect of Warm-Up.** As mentioned in Section 5, the proposed algorithm is vulnerable to noise at early steps, and stale gradients may affect convergence. We compare the convergence of training loss when the warm-up step is selected from $\{0, 50, 500, 5000\}$. As illustrated in the left of Figure 5, we observe that the algorithm may diverge if there is no warm-up at the early stages of training.

**The Effect of Repeatable Dropout.** We also find that the randomness in the dropout layer affects the convergence of the proposed algorithm. In the right of Figure 5, we evaluate the effectiveness of dropout. It is clear that the convergence is affected if there is no repeatable dropout.

## 7   Conclusions

We have considered accelerating the training of Transformer-based language models, and have introduced a novel "Ouroboros" algorithm. We prove Ouroboros is guaranteed to converge to critical points for non-convex problems, and has a similar convergence rate as normal SGD. We conduct experiments on training Transformer-based language models, and experimental results verify that the proposed algorithm can yield a significant speedup without loss of accuracy.

### Acknowledgments

This research was supported in part by DARPA, DOE, NIH, ONR and NSF.

## Footnotes

[2]https://github.com/kimiyoung/transformer-xl/tree/master/pytorch

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
