[Supplementary Material · appendix.pdf]

# A  Algorithm

 In Algorithm 2, we also apply Ouroboros to Adam, an adaptive variant of SGD. We update first
 moment vectors and second moment vectors using gradients computed by the Ouroboros algorithm
 so that the updates in modules can be parallelized.

---

**Algorithm 2** Ouroboros + Adam

---

**Require:**
  Initial weights $w^0 = [w^0_{\mathcal{G}(1)}, ..., w^0_{\mathcal{G}(K)}]$;
  Initial word embedding $V_i^0 = V_o^0$;
  Stepsize: $\gamma$; Small constant $\epsilon = 10^{-8}$;
  Exponential decay: $\beta_1 = 0.9$, $\beta_2 = 0.999$ ;
  $1_{st}$ moment vector: $m^0_{\mathcal{G}(k)} = 0, \forall k, m^0_V = 0$;
  $2_{nd}$ moment vector: $v^0_{\mathcal{G}(k)} = 0, \forall k, v^0_V = 0$;

1: **for** $t = 0, 1, 2, \ldots, T-1$ **do**
2:     **for** $k = 1, \ldots, K$ **in parallel do**
3:         Compute delayed gradient $g_k^t$ for module $k$ following (8):
4:         Compute mixed gradient $g_V^t$ for embedding layer following (9):
5:         Update biased first moment estimate:
           $m^{t+1}_{\mathcal{G}(k)} = \beta_1 \cdot m^t_{\mathcal{G}(k)} + (1-\beta_1) \cdot g_k^t$;
           $m^{t+1}_V = \beta_1 \cdot m^t_V + (1-\beta_1) \cdot g_V^t$;
6:         Update biased second moment estimate:
           $v^{t+1}_{\mathcal{G}(k)} = \beta_2 \cdot v^t_{\mathcal{G}(k)} + (1-\beta_2) \cdot (g_k^t)^2$;
           $v^{t+1}_V = \beta_2 \cdot v^t_V + (1-\beta_2) \cdot (g_V^t)^2$;
7:         Compute bias-correct first moment estimate:
           $\hat{m}^{t+1}_{\mathcal{G}(k)} = m^{t+1}_{\mathcal{G}(k)}/(1-\beta_1^{t+1})$;
           $\hat{m}^{t+1}_V = m^{t+1}_V/(1-\beta_1^{t+1})$;
8:         Compute bias-correct second moment estimate:
           $\hat{v}^{t+1}_{\mathcal{G}(k)} = v^{t+1}_{\mathcal{G}(k)}/(1-\beta_2^{t+1})$;
           $\hat{v}^{t+1}_V = v^{t+1}_V/(1-\beta_2^{t+1})$;
9:         Update weights and embedding layer following Adam:
           $w^{t+1}_{\mathcal{G}(k)} = w^t_{\mathcal{G}(k)} - \gamma \cdot \frac{\hat{m}^{t+1}_{\mathcal{G}(k)}}{\left(\sqrt{\hat{v}^{t+1}_{\mathcal{G}(k)}}+\epsilon\right)}$;
           $V_i^{t+1} = V_o^{t+1} = V_i^t - \gamma \cdot \frac{\hat{m}^{t+1}_V}{\left(\sqrt{\hat{v}^{t+1}_V}+\epsilon\right)}$;
10:     **end for**
11: **end for**
12: Output $w^s$, $V_i^s$ and $V_o^s$ randomly from $\{w^t\}_{t=0}^{T-1}$, $\{V_i^t\}_{t=0}^{T-1}$ and $\{V_o^t\}_{t=0}^{T-1}$.

---

 # B  Proof

 **Proof to Lemma 1**

 *Proof:* Let $\tilde{w} = [V, w]$, it is satisfied that:

$$\nabla f(\tilde{w}^t) = \sum_{k=1}^K \nabla f_{\mathcal{G}(k)}\left(\tilde{w}^t\right) + \nabla f_{V_i}\left(\tilde{w}^t\right) + \nabla f_{V_o}\left(\tilde{w}^t\right) \tag{18}$$

 According to Assumption 1, the following inequality holds that:

$$f(\tilde{w}^{t+1}) \leq f(\tilde{w}^t) + \nabla f(\tilde{w}^t)^T \left(\tilde{w}^{t+1} - \tilde{w}^t\right) + \frac{L}{2}\left\|\tilde{w}^{t+1} - \tilde{w}^t\right\|_2^2. \tag{19}$$

From the update rule in Algorithm 1, we take expectation on both sides and obtain: small

$$\mathbb{E}\left[f(\tilde{w}^{t+1})\right]$$

$$\leq \quad f(\tilde{w}^t) - \gamma_t \nabla f(\tilde{w}^t)^T \Big( \sum_{k=1}^{K} \nabla f_{\mathcal{G}(k)} \left(\tilde{w}^{t-K+k}\right) + \frac{1}{2}\nabla f_V \left(\tilde{w}^{t-K+1}\right) + \frac{1}{2}\nabla f_V \left(\tilde{w}^t\right)$$

$$+ \nabla f\left(\tilde{w}^t\right) - \nabla f\left(\tilde{w}^t\right) \Big) + \frac{L\gamma_t^2}{2}\mathbb{E}\Big\| \sum_{k=1}^{K} \nabla f_{\mathcal{G}(k), x_{i(t-K+k)}} \left(\tilde{w}^{t-K+k}\right)$$

$$+ \frac{1}{2}\nabla f_{V, x_{i(t-K+1)}} \left(\tilde{w}^{t-K+1}\right) + \frac{1}{2}\nabla f_{V, x_{i(t)}} \left(\tilde{w}^t\right) - \nabla f(\tilde{w}^t) + \nabla f(\tilde{w}^t) \Big\|_2^2$$

$$= \quad f(\tilde{w}^t) - \left(\gamma_t - \frac{L\gamma_t^2}{2}\right) \left\|\nabla f(\tilde{w}^t)\right\|_2^2 + \frac{L\gamma_t^2}{2}\mathbb{E}\Big\| \sum_{k=1}^{K} \nabla f_{\mathcal{G}(k), x_{i(t-K+k)}} \left(\tilde{w}^{t-K+k}\right)$$

$$+ \frac{1}{2}\nabla f_{V, x_{i(t-K+1)}} \left(\tilde{w}^{t-K+1}\right) + \frac{1}{2}\nabla f_{V, x_{i(t)}} \left(\tilde{w}^t\right) - \nabla f(\tilde{w}^t) \Big\|_2^2$$

$$- \left(\gamma_t - L\gamma_t^2\right) \nabla f(\tilde{w}^t)^T \Big( \sum_{k=1}^{K} \nabla f_{\mathcal{G}(k)} \left(\tilde{w}^{t-K+k}\right) + \frac{1}{2}\nabla f_V \left(\tilde{w}^{t-K+1}\right) + \frac{1}{2}\nabla f_V \left(\tilde{w}^t\right) - \nabla f(\tilde{w}^t) \Big)$$

$$= \quad f(\tilde{w}^t) - \left(\gamma_t - \frac{L\gamma_t^2}{2}\right) \left\|\nabla f(\tilde{w}^t)\right\|_2^2 + \underbrace{\frac{L\gamma_t^2}{2}\mathbb{E}\Big\| \sum_{k=1}^{K} \nabla f_{\mathcal{G}(k), x_{i(t-K+k)}} \left(\tilde{w}^{t-K+k}\right) - \sum_{k=1}^{K} \nabla f_{\mathcal{G}(k)}(\tilde{w}^t) \Big\|_2^2}_{Q_1}$$

$$+ \underbrace{\frac{L\gamma_t^2}{2}\mathbb{E}\Big\| \frac{1}{2}\nabla f_{V, x_{i(t-K+1)}} \left(\tilde{w}^{t-K+1}\right) + \frac{1}{2}\nabla f_{V, x_{i(t)}} \left(\tilde{w}^t\right) - \nabla f_V(\tilde{w}^t) \Big\|_2^2}_{Q_2} \qquad (20)$$

$$- \underbrace{\left(\gamma_t - L\gamma_t^2\right) \nabla f(\tilde{w}^t)^T \Big( \sum_{k=1}^{K} \nabla f_{\mathcal{G}(k)} \left(\tilde{w}^{t-K+k}\right) + \frac{1}{2}\nabla f_V \left(\tilde{w}^{t-K+1}\right) + \frac{1}{2}\nabla f_V \left(\tilde{w}^t\right) - \nabla f(\tilde{w}^t) \Big)}_{Q_3}$$

where the equalities follow from the unbiased gradient $\mathbb{E}\left[\nabla f_{x_i}(w)\right] = \nabla f(w)$ and $[\nabla f_{\mathcal{G}(k)}(w)]_j = 0$, $\forall j \notin \mathcal{G}(k)$. Because of $\|x + y\|_2^2 \leq 2\|x\|_2^2 + 2\|y\|_2^2$, we have the upper bound of $Q_1$ as follows:

$$Q_1$$

$$= \quad \frac{L\gamma_t^2}{2}\mathbb{E}\Big\| \sum_{k=1}^{K} \nabla f_{\mathcal{G}(k), x_{i(t-K+k)}}(\tilde{w}^{t-K+k}) - \sum_{k=1}^{K} \nabla f_{\mathcal{G}(k)}(\tilde{w}^t) - \sum_{k=1}^{K} \nabla f_{\mathcal{G}(k)}(\tilde{w}^{t-K+k}) + \sum_{k=1}^{K} \nabla f_{\mathcal{G}(k)}(\tilde{w}^{t-K+k}) \Big\|_2^2$$

$$\leq \quad L\gamma_t^2 \mathbb{E}\underbrace{\Big\| \sum_{k=1}^{K} \nabla f_{\mathcal{G}(k), x_{i(t-K+k)}}(\tilde{w}^{t-K+k}) - \sum_{k=1}^{K} \nabla f_{\mathcal{G}(k)}(\tilde{w}^{t-K+k}) \Big\|_2^2}_{Q_4}$$

$$+ L\gamma_t^2 \underbrace{\Big\| \sum_{k=1}^{K} \nabla f_{\mathcal{G}(k)}(\tilde{w}^{t-K+k}) - \sum_{k=1}^{K} \nabla f_{\mathcal{G}(k)}(\tilde{w}^t) \Big\|_2^2}_{Q_5}. \qquad (21)$$

Similarly, we get the upper bound of $Q_2$ as follows:

$$Q_2 \quad \leq \quad \frac{L\gamma_t^2}{4}\mathbb{E}\left\|\nabla f_{V, x_{i(t-K+k)}}(\tilde{w}^{t-K+1}) - \nabla f_V(\tilde{w}^t)\right\|_2^2 + \frac{L\gamma_t^2}{4}\left\|\nabla f_{V, x_{i(t)}}(\tilde{w}^t) - \nabla f_V(\tilde{w}^t)\right\|_2^2$$

$$\leq \quad \frac{L\gamma_t^2}{2}\underbrace{\mathbb{E}\left\|\nabla f_{V, x_{i(t-K+k)}}(\tilde{w}^{t-K+1}) - \nabla f_V(\tilde{w}^{t-K+1})\right\|_2^2}_{Q_6} + \frac{L\gamma_t^2}{2}\underbrace{\mathbb{E}\left\|\nabla f_V(\tilde{w}^{t-K+1}) - \nabla f_V(\tilde{w}^t)\right\|_2^2}_{Q_7}$$

$$+ \frac{L\gamma_t^2}{4}\left\|\nabla f_{V, x_{i(t)}}(\tilde{w}^t) - \nabla f_V(\tilde{w}^t)\right\|_2^2. \qquad (22)$$

Because of $xy \leq \frac{1}{2}\|x\|_2^2 + \frac{1}{2}\|y\|_2^2$, we have:

$$Q_3 \leq \frac{\gamma_t - L\gamma_t^2}{2}\left\|\nabla f(\tilde{w}^t)\right\|_2^2 + \frac{\gamma_t - L\gamma_t^2}{2}\left\|\sum_{k=1}^{K}\nabla f_{\mathcal{G}(k)}\left(\tilde{w}^{t-K+k}\right) - \sum_{k=1}^{K}\nabla f_{\mathcal{G}(k)}\left(\tilde{w}^t\right)\right\|_2^2$$
$$+ \frac{\gamma_t - L\gamma_t^2}{8}\left\|\nabla f_V\left(\tilde{w}^{t-K+1}\right) - \nabla f_V(\tilde{w}^t)\right\|_2^2. \tag{23}$$

According to Assumption 2, we can bound $Q_4$ as follows:

$$Q_4 = \sum_{k=1}^{K}\mathbb{E}\left\|\nabla f_{\mathcal{G}(k),x_{i(t-K+k)}}(\tilde{w}^{t-K+k}) - \nabla f_{\mathcal{G}(k)}(\tilde{w}^{t-K+k})\right\|_2^2$$
$$\leq \sum_{k=1}^{K}\mathbb{E}\left\|\nabla f_{\mathcal{G}(k),x_{i(t-K+k)}}(\tilde{w}^{t-K+k})\right\|_2^2$$
$$\leq KM, \tag{24}$$

where the first equality follows from the definition of $\nabla f_{\mathcal{G}(k)}(w)$ so that $[\nabla f_{\mathcal{G}(k)}(w)]_j = 0$, $\forall j \notin \mathcal{G}(k)$ and the last inequality is from Assumption 2. Similarly, we can also bound $Q_6$ as follows:

$$Q_6 \leq M. \tag{25}$$

We can get the upper bound of $Q_5$:

$$Q_5 = \sum_{k=1}^{K}\left\|\nabla f_{\mathcal{G}(k)}(\tilde{w}^{t-K+k}) - \nabla f_{\mathcal{G}(k)}(\tilde{w}^t)\right\|_2^2$$
$$\leq \sum_{k=1}^{K}\left\|\nabla f(\tilde{w}^{t-K+k}) - \nabla f(\tilde{w}^t)\right\|_2^2$$
$$\leq L^2\sum_{k=1}^{K}\left\|\sum_{j=\max\{0,t-K+k\}}^{t-1}\left(\tilde{w}^{j+1} - \tilde{w}^j\right)\right\|^2$$
$$\leq L^2\gamma_{\max\{0,t-K+1\}}^2 K\sum_{k=1}^{K}\sum_{j=\max\{0,t-K+k\}}^{t-1}\left\|\sum_{k=1}^{K}\nabla f_{\mathcal{G}(k),x_{(j-K+k)}}\left(\tilde{w}^{j-K+k}\right)\right.$$
$$\left. + \nabla f_{V,x_{(j-K+k)}}\left(\tilde{w}^{j-K+k}\right) + \nabla f_{V,x_{(j)}}\left(\tilde{w}^j\right)\right\|_2^2$$
$$\leq KL\gamma_t\frac{\gamma_{\max\{0,t-K+1\}}}{\gamma_t}\sum_{k=1}^{K}\sum_{j=\max\{0,t-K+k\}}^{t-1}\left\|\sum_{k=1}^{K}\nabla f_{\mathcal{G}(k),x_{(j-K+k)}}\left(\tilde{w}^{j-K+k}\right)\right.$$
$$\left. + \nabla f_{V,x_{(j-K+k)}}\left(\tilde{w}^{j-K+k}\right) + \nabla f_{V,x_{(j)}}\left(\tilde{w}^j\right)\right\|_2^2$$
$$\leq L\gamma_t\sigma K^3(K+4)M, \tag{26}$$

where the second inequality is from Assumption 1, the fourth inequality follows from that $L\gamma_t \leq 1$ and the last inequality follows from $\|z_1 + ... + z_r\|_2^2 \leq r(\|z_1\|_2^2 + ... + \|z_r\|_2^2)$, Assumption 2 and $\sigma := \max_t \frac{\gamma_{\max\{0,t-K+1\}}}{\gamma_t}$. Similarly, we can bound $Q_7$:

$$Q_7 \leq L\gamma_t\sigma K^2(K+4)M. \tag{27}$$

Integrating the upper bound of $Q_1$ to $Q_7$ in (21), we have:

$$\mathbb{E}\left[f(\tilde{w}^{t+1})\right] - f(\tilde{w}^t) \leq -\frac{\gamma_t}{2}\left\|\nabla f(\tilde{w}^t)\right\|^2 + \gamma_t^2 LM_K, \tag{28}$$

where we let $M_K = (K + \frac{3}{4})M + \sigma(\frac{K^2}{2} + K^3)(K+4)M$.

□

**Proof to Theorem 1**

*Proof:* When $\gamma_t$ is constant and $\gamma_t = \gamma$, we have $\sigma = 1$. Because of the definition of $M_K$ and taking total expectation of (15) in Lemma 1, we obtain:

$$\mathbb{E}\left[f(w^{t+1})\right] - f(w^t) \leq -\frac{\gamma}{2}\mathbb{E}\left\|\nabla f(w^t)\right\|_2^2 + \gamma^2 L M_K. \tag{29}$$

Summing (29) from $t = 0$ to $T - 1$, we have:

$$\begin{aligned}
\mathbb{E}\left[f(w^T)\right] - f(w^0) \leq\ & -\frac{\gamma}{2}\sum_{t=0}^{T-1}\mathbb{E}\left\|\nabla f(w^t)\right\|_2^2 \\
& + T\gamma^2 L M_K.
\end{aligned} \tag{30}$$

Supposing $w^*$ is the optimal solution for $f(w)$, it holds that $f(w^*) - f(w^0) \leq \mathbb{E}\left[f(w^T)\right] - f(w^0)$. Rearranging inequality (30) and dividing both sides by $\frac{2T}{\gamma}$, we complete the proof.

$\square$

**Proof to Theorem 2**

*Proof:* $\{\gamma_t\}$ is a diminishing sequence and $\gamma_t = \frac{\gamma_0}{1+t}$, such that $\sigma \leq K$ and $M_K = (K + \frac{3}{4})M + (\frac{K^3}{2} + K^4)(K + 4)M$. Taking total expectation of (15) in Lemma 1 and summing it from $t = 0$ to $T - 1$, we obtain:

$$\begin{aligned}
\mathbb{E}\left[f(w^T)\right] - f(w^0) \leq\ & -\frac{1}{2}\sum_{t=0}^{T-1}\gamma_t\mathbb{E}\left\|\nabla f(w^t)\right\|_2^2 \\
& + \sum_{t=0}^{T-1}\gamma_t^2 L M_K.
\end{aligned} \tag{31}$$

Suppose $w^*$ is the optimal solution for $f(w)$; therefore $f(w^*) - f(w^0) \leq \mathbb{E}\left[f(w^T)\right] - f(w^0)$. Rearranging inequality (31) and dividing both sides by $T$, we complete the proof.

$\square$