[Reviews · NeurIPS 2019]

Reviewer 1



Authors propose the model-parallel gradient descent algorithm for the purposes of speeding up training of Transformer based language models. Overall the paper is well written and experiments are convincing in demonstration of validity of the approach. My main question is whether authors have tried training much larger Transformer models that don't fit into one GPU using their algorithm

Reviewer 2



This paper provides a way to divide very large Transformer-based deep Neural Networks to several modules in an efficient way for training. Transformer-based language models require a huge computational cost and time, therefore model parallelism is required if a model is too large to fit in a single computing device. However, the basic Transformer training requires to wait for the previous layers' gradients to compute the current layer gradients. Therefore, some GPUs becomes idle during training when we split the model into multiple GPUs. The proposed accelerating training method enables Transformer-based language model parallelism by avoiding the backward locking. The novelty of the idea and the contribution of the theoretical analysis is somewhat limited, but overall this paper shows the great progress on the parallelization on Transformer-based Language model training.

Reviewer 3



The paper introduces a new method for model-parallel training, where layers of a model are distributed across multiple accelerators. The method avoids locking in the backward pass by using stale gradients during back-propagation. I'm not aware of any prior work that took such an approach. Furthermore, the authors provide theoretical claims and empirical results to demonstrate that their method has convergence properties similar to conventional SGD, despite using stale gradients. The lack of effective model-parallel training is a major roadblock for scaling up model sizes, and the proposed approach promises to overcome this issue. Minor: 74: the notation grad f_{l_x_{i(t)}} is not used in equation (6). It would also be useful to remind what this notation means next to equation (8). 84: should be "any ... method" 116 or 127: it would be good to say that proofs are in the supplementary material, rather than leaving it unstated whether the authors have proven the key theorems.

[Author Response · NeurIPS 2019]

We thank all the reviewers for their positive and constructive comments.

To Reviewer #1

**Q1**: "My main question is whether authors have tried training much larger Transformer models that don't fit into one GPU using their algorithm."
**A1**: Yes, in our experiment, training 12-layer vanilla Transformer and Transformer-XL models with batch size 22 cannot fit into one GPU. So we need to split them into multiple modules with each module placed on one GPU. Our method also works for models which are so large that even training with batch size 1 cannot fit into one GPU.

To Reviewer #2

**Q1**: "I strongly encourage authors to share the code used in experiments for researchers and practitioners in the future as soon as possible as noted in the paper."
**A1**: Yes, we will release our source code on GitHub to encourage further research.

To Reviewer #3

**Q1**: "74: the notation grad $f_{l,x_{i(t)}}$ is not used in equation (6). It would also be useful to remind what this notation means next to equation (8)."
**A1**: The notation grad $f_{l,x_{i(t)}}$ is trying to clarify the relation between equations (5) and (6). Thanks for your good suggestion and we will also add it close to the equation (8).

**Q2**: "84: should be 'any ... method'."
**A2**: We will correct this typo in the future version.

**Q3**: "116 or 127: it would be good to say that proofs are in the supplementary material, rather than leaving it unstated whether the authors have proven the key theorems."
**A3**: Thank you for the useful suggestion and we will revise it in the updated version.

**Q4**: "Could the authors please comment on why speed gains remain below $2\times$, even as up to $4\times$ the number of GPUs is used? The speed-ups should break down into two parts: the 'time per step' decreases as more GPUs are added, but the 'number of steps to convergence' increases as a function of $K$. The first of these could potentially improve with new hardware and systems software, while the second is inherent to the proposed method. What is the breakdown between these two components?"
**A4**: It is an ideal case to obtain linear speedup, using $K\times$ machines to achieve $K\times$ speedup regarding time. However, it is impossible to achieve even for data parallelism. It is also hard to define the concept of "step" in our model parallelism, because devices compute gradients from different steps in parallel. The goal of our method is to guarantee that there is no idle machines during the training and fully utilize all computing resources. On the contrary, the vanilla backpropagation algorithm is a sequential process and the other devices are idle when one device is processing. Regarding the effect of $K$ on the number of steps to converge, it is true that increasing $K$ may require more steps to converge as the Theorem 1 suggests. A better algorithm which can mitigate the effect of $K$ is one of the potential future direction of this paper.

**Q5**: "Relatedly, I also would be interesting in being able to better extrapolate how the algorithm might behave in the following regimes: (a) A large number of layers (64+) is split across a relatively small number of devices ($\sim$4); and (b) the same 12-layer model used in the experiments is split across 11 devices."
**A5**: For the regime (a), we think the speedup has little relation with the number of layers. Thus, it could get about the same $2\times$ speedup as the 12-layer Transformer. As for the regime (b), due to the limited resources, we validate our algorithm by varying $K$ from 3 to 5 and we didn't test the performance of our method when there are more (e.g., 11) devices. We guess there exists a number that increasing the number of devices will not get further speedup. Finally, we want to emphasize that we can combine data parallelism with our model parallelism to obtain further speedup, which is nontrivial for practitioners.

[Meta-Review · NeurIPS 2019]

This paper studies the problem of parallelising large transformer-based language models. It goes beyond data parallelism in that it focuses on splitting the model when it does not fit in the memory of a single GPU. The idea is to segment the model into groups such that GPUs do not sit around waiting on others to pass gradients ( this is the case for layer-wise parallel solutions where each layer is on its own GPU). The model then allows backpropagation to use stale gradients between groups. An L-layer network is split into K modules so that the weights of the network are divided into K groups and each group is placed on a GPU. In the experiments, Transformer-based language model is split into K modules and allocated sequentially onto K GPUs. Theoretical guarantees for convergence are presented and experiments show (modest) improvements in training speed without task hurting performance. The paper would be a good addition to the conference, there is support for its inclusion in the conference.